



# Surface charge of environmental and radioactive airborne particles
Gyoung G. Jang[1,*], Alexander I. Wiechert[2,*], Austin P. Ladshaw[1], Tyler Spano[1], Joanna McFarlane[1],
Kristian Myhre[1], Sotira Yiacoumi[2], Costas Tsouris[1,2]
[1] Oak Ridge National Laboratory, Oak Ridge, TN 37831-6181, United States
[2] Georgia Institute of Technology, Atlanta, GA 30332-0373, United States
These authors contributed equally to this work.*
*Correspondence to*: Costas Tsouris (tsourisc@ornl.gov)
**Abstract.** Self-charging of radioactive uranium oxide particles was measured by comparing the electrostatic surface-charge
characteristics of the uranium particles to various airborne dust particulates. Though radioactive aerosols can gain charge through
various decay mechanisms, researchers have traditionally assumed that the radioactive aerosols do not carry any additional charge
relative to other atmospheric dust particles as a consequence of charge neutralization over time. In this work, we evaluate this
assumption by directly examining the surface charge and charge density on airborne uranium oxide particles and then comparing
those characteristics with charging of other natural and engineered airborne dust particles. Based on electric field–assisted particle
levitation in air, the surface charge, charge distribution as a function of particle size, and surface charge density were determined
for uranium oxide aerosols (<1 μm) and other nonradioactive dusts, including urban dust, Arizona desert dust, hydrophilic and
hydrophobic silica nanoparticles, and graphene oxide powders. Of these dusts, uranium oxide aerosols exhibited the highest surface
change density. Additionally, a self-charging model was employed to predict average charge gained from radioactive decay as a
function of time. The experimental and theoretical results suggest that radioactive self-charging likely occurs on airborne particles
containing radionuclides and may potentially affect the transport of radioactive particles in the atmosphere.
**Copyright statement**
## 1. Introduction
Considering the substantial public health risks and environmental damage that could arise from the deposition of debris from
unwanted nuclear events, the development of tools that can accurately predict the transport of such a debris is of great importance
(Pöllänen et al. 1997, Yamauchi 2012, Draxler et al., 2015, Yoshikane et al., 2016). To effectively predict the transport of
radioactive particles in the atmosphere, one must first obtain a reliable estimation of the aerosol's surface charge characteristics
(ApSimon et al., 1989; Lee et al., 1995; Pöllänen et al., 1997; Andrews et al., 2020). Even for nonradioactive particles, the
electrostatic forces acting on airborne particulates play a significant role in the transport of dusts that are lofted into the atmosphere
and transported thousands of kilometers from their point of origin (Kok et al., 2006; Kok et al., 2008). Heavy charging of airborne
dusts during dust storms is well documented in the literature; some storms develop an electrical field in excess of 100 kV/m
(Gensdarmes et al. 2001). These electric fields arise from contact charging of wind-blown dust particles as they collide with one
another and transfer charge though the triboelectric effect. Triboelectric charging occurs even when all particles comprise the same
material, with some particles becoming positively charged while others are negatively charged. Therefore, particle characteristics



such as size, density, and chemical properties play a key role in atmospheric transportation. When compared with other airborne
dust particles, particulates that contain radionuclides can, in addition to triboelectric charging, be self-charged through radioactive
decay (Gensdarmes et al. 2001). The impact of radioactive self-charging depends on the activity (i.e., decay incidents per unit of
time) of the radioisotopes found in the aerosol and on the type of radioactive decay (i.e., α or β decay) that those isotopes undergo.
For example, the charge gained from each incident of α-decay depends on the number of free electrons released when the helium
nucleus emitted during decay emerges from the particle (Figure 1). Additionally, the number of free electrons released primarily
depends upon the energy of the α-particle and the characteristics of the medium through which it passes.

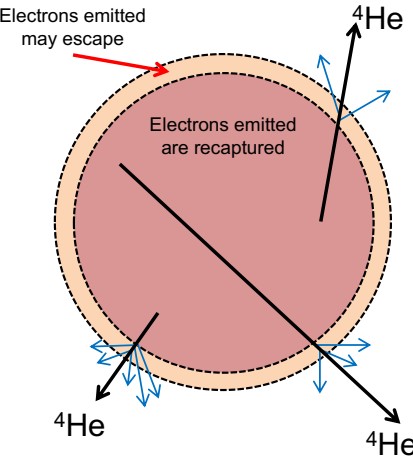

**Figure 1: Schematic demonstrating aerosol charging from alpha decay.**
Despite the unique nature of this self-charging behavior, researchers have traditionally assumed that radioactive particles have
similar charging characteristics to those of other aerosols in the atmosphere, meaning that the electrostatic surface interactions
from radioactive decay can be neglected (Seinfeld et al., 2006). One generally accepted hypothesis is that ionizing radiation induces
charge neutralization for particles in the atmosphere and, therefore, the charging attributable to radioactive decay will ultimately
be negligible (Greenfield, 1956; Greenfield, 1957). Recently, however, some studies have reported that radioactivity could induce
strong self-charging, which leads to an accumulation of charge on both the radioactive particles and background aerosols, thus
significantly changing the electrical properties of the local atmosphere (Kim et al., 2015). Additionally, although radioactive
airborne particles typically contain a mixture of radionuclides and other background dust particles, little reporting has directly
compared particle charge on radioactive aerosols with the charge measured on other dust particle under the same conditions.
Here, we investigate the electrostatic charge on various radioactive, natural, and engineered dust particles using an electrodynamic
balance (similar to Millikan's oil drop apparatus; Millikan, 1911) to determine the particles' surface charge, charge distribution as
a function of particle size, and surface charge density. These measurements are based on direct observation of dust levitation in a
known electric field to resolve the elementary charge on each particle dispersed in air. Our experiments are reminiscent of
Millikan's famous experiments with oil drops in air, which demonstrated the discrete nature of electric charge (Millikan, 1911).
Radioactive uranium oxide, urban dust, Arizona desert dust, hydrophilic silica, hydrophobic silica, and graphene oxide particles
were examined, and their charging characteristics were compared to better understand the impact of the radioactive self-charge
behavior. Radioactive decay simulations were performed to obtain an approximation of the self-charging rate and its potential
impact on self-charging of the uranium oxide particles examined in this study.



**2. Methodology**
**2.1 Materials**
High–surface area graphene oxide (N002-PDE: C = 60–80 at. % and O = 10–30 at. %) was purchased from Angstron Materials.
Hydrophobic (Aerosil R8200) and hydrophilic (Aerosil 200) fumed silica ($SiO_2$) nanoparticles with specific surface areas of 135–
185 and 175–225 $m^2$/g, respectively, were procured from Evonik Industries. Arizona dust with a size distribution of 0.97–352 μm
(ISO 12103-1, A4 Coarse Grade) was obtained from Powder Technology, Incorporated. Additionally, NIST SRM 1649b urban
dust was acquired from Sigma-Aldrich.
To analyze radioactive self-charging, uranium oxide ($UO_2$) particles were prepared by reducing $U_3O_8$ in a mixture of 4 mol % $H_2$
in argon at 600°C for several hours. Following an assay of the $UO_2$ particles, we found that the uranium content of the particles
was 99.78 mol % $^{238}U$, 0.22 mol % $^{235}U$, and 0.0054 mol % $^{234}U$. The $UO_2$ powder, initially prepared though $U_3O_8$ reduction, was
estimated to have an average particle size of 20 μm, which was far larger than the 1 μm desired for our investigation of airborne
particulates. Therefore, the size of the $UO_2$ particles was reduced to the desired diameter by grinding in a ball mill with 5 mm
diameter $ZrO_2$ beads and in an agate mortar. The samples were ground in a slurry of anhydrous ethanol to help minimize the release
of minute airborne $UO_2$ particles. Liquid fractionation, separating particles by sedimentation rate in a fluid, was then performed in
columns containing anhydrous ethanol or deionized water (ASTM D422-63; Standard Tests Method for Particle-Size Analysis of
Soils). Two suspensions were prepared using this method, including a suspension of submicron $UO_2$ particles and a suspension of
$UO_2$ particles between 1 and 1.5 μm in diameter.
**2.2 Electrical charge measurement**
An EX-9929A electrodynamic balance purchased from PASCO was used as part of the experimental setup (Figure 2) for analyzing
the charge of airborne dust particles. The electrical charge carried by a particle can be determined by measuring the particle's
velocity in a known electric field referenced against the velocity of the same particle in free fall in the absence of an electric field.
This methodology was demonstrated in the traditional Millikan oil drop experiment, which analyzed the behavior of small charged
oil droplets weighting $10^{-12}$ g or less. The droplet mass can be calculated using Stokes' law by first measuring the free-fall velocity
of the droplet in air. Then, by observing the velocity of the same droplet as it rises in a known electric field, the force needed to
levitate the droplet can be determined and the charge carried by the droplet can be calculated (Figure 2).

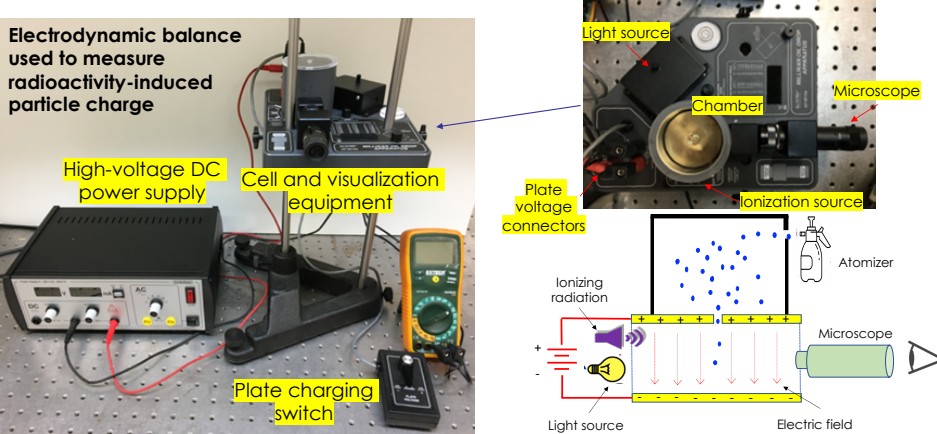

**Figure 2: Experimental setup for the electrodynamic balance used to observe the behavior of particles in an electric field and determine**
**their charge through a force balance.**





**2.3 Characterization of materials**
Scanning electron microscopy (SEM) and energy-dispersive x-ray spectroscopy were conducted using a field emission scanning
electron microanalyzer (Merlin, Carl Zeiss AG). Fourier-transform infrared spectroscopy was performed in attenuated total
reflectance mode with 32 scans using a Perkin Elmer Frontier instrument. Powder x-ray diffraction data were collected using a
Proto AXRD benchtop diffractometer in Bragg-Brentano configuration with a Cu Kα source, λ=1.5406 Å and Mythen 1K-1D
detector. A 0.2 mm divergence slit and incident and diffracted beam Soller slits were employed to reduce axial divergence. Scans
were performed along a 2θ range of 10° to 60° with a step velocity of 1.2° 2θ/min. Samples of uranium, urban, and desert dust
particles were identified using the ICDD powder diffraction file (PDF4+). GSASII was used to determine $U_3O_8$ and $UO_2$ phase
fractions. CIF files for the two phases were imported, and a light refinement of the lattice parameters was performed. Next,
background refinements were carried out using Chebyshev polynomial order corrections with 12 coefficients. Particle size,
microstrain, and phase fraction refinements were then performed with a resulting $R_{wp}$ = 11.891%. From this analysis, $UO_2$ and
$U_3O_8$ phase fractions were found to be 0.8819 and 0.1181, respectively.
**2.4 Self-charging modeling methodology**
To assess the potential impact of radioactive self-charging on the mean charge of the $UO_2$ particles examined in this study, a
simplified charge balance (Eq. 1) was used to estimate the charging rate for g-sized particles ($dJ_g/dt$) solely as a function of the
particle's activity. The activity of the particle is then derived from the sum of the individual activities of each radioisotope present
in the particle. For the $i^{th}$ isotope present in the particle, the charging rate is determined from the isotope's activity ($A_i$) and the
average charge gained per decay incident involving the $i^{th}$ isotope ($C_i$). Furthermore, the activity of the $i^{th}$ isotope depends upon
both the isotope's number concentration ($Z_i$) and decay rate ($\lambda_i$). Since the emission of γ radiation will not affect the particle's
charge, we can, from the perspective of self-charging, consider α and β⁻ decay to be the only notable modes of radioactive decay
in this study. Thus, the value of $C_i$ can be calculated using Eq. 2 where $b_{\alpha,i}$ and $b_{\beta,i}$ are the branch fractions of the $i^{th}$ isotope for α
and β decays, respectively, and $M_\alpha$ is the net charge gained form each incident of α decay. Since the value of $M_\alpha$ cannot be reliably
predicted using analytical methods, one must generally rely on experiments to determine the $M_\alpha$ average value (Gensdarmes et al.,
2001). Unfortunately, the necessary experiments have not been performed using $UO_2$ to determine either the approximate range of
values for $M_\alpha$ or its average value. Based on previous experiments with other materials, however, we can reasonably assume that
the mean value of $M_\alpha$ likely falls somewhere between 6 and 12 (Anno et al., 1963; Clement et al., 1992).
$$\frac{dJ_g}{dt} = \sum C_i A_i = \sum C_i \lambda_i Z_i \qquad (1)$$
$$C_i = b_{\beta,i} + M_\alpha b_{\alpha,i} \qquad (2)$$
The number of concentrations for each of the radioisotopes in the particle was determined as a function of time using decay chain
simulations performed with the Implicit Branching Isotope System (IBIS) module of the Ecosystem Software developed at the
Georgia Institute of Technology (Ladshaw et al., 2015). Starting with an initial set of known isotopes, IBIS automatically generates
a full list of the nuclides formed from radioactive decay of the initial isotopes based on decay data for registered nuclides in the
Nuclide Wallet Cards distributed by Brookhaven National Laboratory (Tuli, 2011). Decay rate constants and branch fractions for
each nuclide are also drawn from the Nuclide Wallet Cards. This list of nuclides is then sorted to guarantee that each parent isotope
will always have a lower index number than its decay by-products. As described by Ladshaw et al. (2020), the nuclides must be
sorted in this fashion to ensure that decay with any number of branches can be solved using the analytical closed-form solution





given in Eq. 3. Additionally, $v_i^k$ (Eq. 4) and $u_i^k$ (Eq. 5) represent the $i^{th}$ row element of the $k^{th}$ eigenvector in the eigenvector matrix
and inverse eigenvector matrix, respectively.

$$Z_i^t = \sum_{j \leq i} \left( \sum_{k=j}^{i} v_i^k u_k^j e^{-\lambda_k t} \right) Z_j^0 \tag{3}$$

$$v_i^k = \begin{cases} 0 & \text{for} \quad i < k \\ 1 & \text{for} \quad i = k \\ -\dfrac{1}{\lambda_k - \lambda_i} \displaystyle\sum_{j<i} v_j^k b_{ij} \lambda_j & \text{for} \quad i > k \end{cases} \tag{4}$$

$$u_i^k = \begin{cases} 0 & \text{for} \quad i < k \\ 1 & \text{for} \quad i = k \\ -\displaystyle\sum_{j<i} u_j^k v_i^j & \text{for} \quad i > k \end{cases} \tag{5}$$

**3. Results and discussion**
**3.1 Experimental results**
Airborne $UO_2$ particles that have a diameter of ~1 μm rising in an electric field with an applied potential of 300 V are shown in
Figure 3. All the particles examined in this study were blown directly into the chamber of the electrodynamic balance apparatus
using a 3 mL plastic pipette dispenser. Before blowing the particles into the balance chamber, the pipette was vigorously shaken
for 1 min to induce charging though the triboelectric effect. Immediately after the particles were injected into the cell, an electric
field was applied to disperse and control the motion of the particles in air. From among the particles that were in view, a number
of particles were selected, and their velocities were monitored under various electric fields to determine their approximate size,
density, and charge.

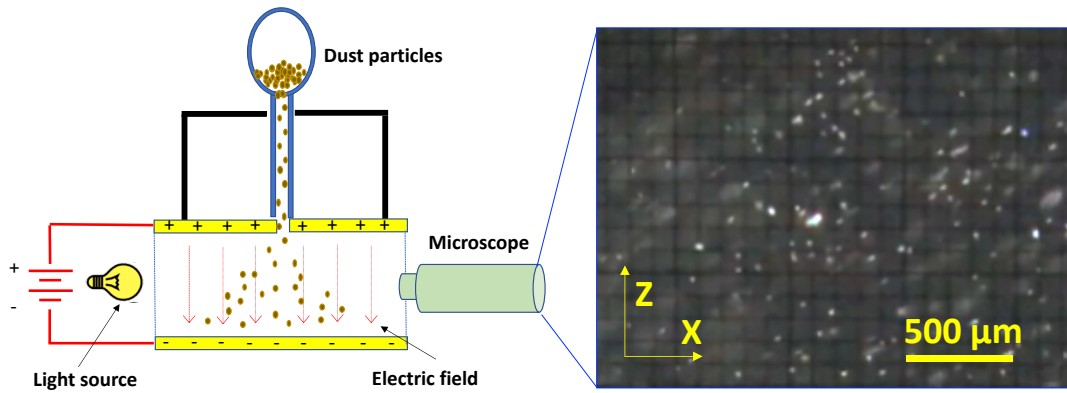


**Figure 3: Experimental observation of airborne radioactive particles dispersed in the balance chamber and rising in an electric field**
**with an applied potential.**





The electric charge ($q$) carried by a particle can be determined using the formula given in Eq. 6. In this formula, $v_f$ is the velocity
of the particle in free fall (m/s), $v_r$ is the upward velocity of the particle in a known electric field, $g$ is the gravitational acceleration
(m/s$^2$), $\rho$ is the particle density (kg/m$^3$), $d$ is the distance separating the plates (m), $V$ is the potential difference across the plates
(V), $a$ is the particle's radius (m), $b$ is a constant equal to $8.2 \times 10^{-3}$ Pa • m, $p$ is the atmospheric pressure (Pa), and $\eta$ is the viscosity
of air (N • s/m$^2$). A more detailed description of Eq. 6 is available in the Supplementary Information.

$$q = \frac{4\pi d\rho g(v_f+v_r)}{3(Vv_f)}\left[\frac{9\eta v_f}{2\rho g}\left(\frac{1}{1+\frac{b}{pa}}\right)\right]^{3/2} \tag{6}$$

A combination of SEM imaging and free-fall particle experiments were performed to provide an approximation of the average
particle size for each of the materials examined in this study. SEM images showing the size and morphology of the UO$_2$, urban
dust, Arizona dust, hydrophilic SiO$_2$, hydrophobic SiO$_2$, and graphene oxide particles are shown in Figure 4. The size of the UO$_2$
particles observed was in the range of 0.3 to 1.5 μm. Urban and Arizona dust particles observed through SEM were of a somewhat
larger size range, between 0.1 and 5.0 μm in diameter. Larger crystalline silica agglomerates, fibers, and other debris were also
observed in both dusts but were not considered for the purposes of this study on the basis of their size (100s of microns) and relative
rarity. Most of these large particles are expected to settle rapidly, leaving only the finer particles observable in the controlled
electric field (100–500 V). Engineered hydrophilic and hydrophobic SiO$_2$ nanoparticles (<10 nm) aggregate into agglomerates that
are several microns in diameter. High–surface-area graphene oxide sheets were also chosen for their high theoretical conductivity
(e.g., $10^7$–$10^8$ S/m), which allows them to be easily controlled in a given electric field. The graphene layers were agglomerated to
form spherical particles with diameters of 5–40 μm.

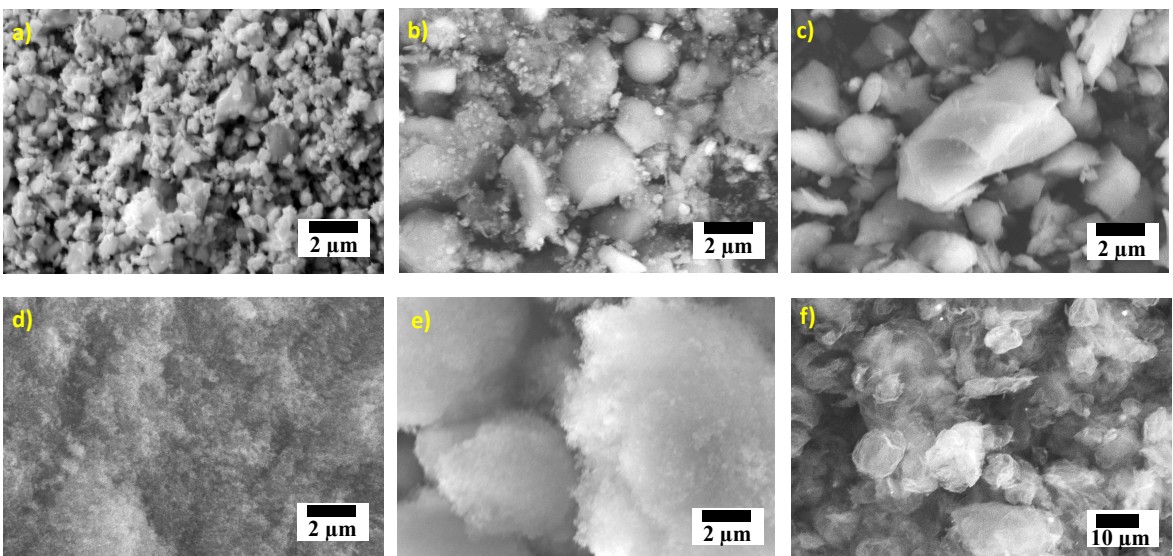

**Figure 4: SEM images of various air borne and engineered particles; (a) uranium oxide particles, (b) urban dust, (c) Arizona dust,**
**(d) hydrophilic SiO$_2$, (e) hydrophobic SiO$_2$, and (f) graphene oxide powder.**
Each particle class was also examined in free-fall mode to provide a more complete understanding of the particle size that can be
used when analyzing the levitation experiments. The physical and material characteristics of the particles are summarized in
Table 1. The uranium oxide particles examined are a mixture of UO$_2$ and U$_3$O$_8$ formed through the incomplete oxidation of U$_3$O$_8$.





Oxidation is known to be an incomplete process based on findings from Raman spectroscopy (see Supplementary Information),
which shows a residual concentration of $U_3O_8$ on the particle surface. Additionally, x-ray diffraction indicated that the $UO_2$ ($\rho = 8.3$
$g/cm^3$) and $U_3O_8$ ($\rho = 10.97$ $g/cm^3$) phase fractions were 0.8819 and 0.1181, respectively (Supplementary Information). Based on
the arithmetic mean of these values, the stoichiometry of the oxide particle was determined to be $UO_{2.191}$ with an approximate
density of 10.65 $g/cm^3$. Based on this density and the terminal velocity of the $UO_2$ particles in free fall, the average size of the $UO_2$
particles was estimated to be $0.48 \pm 0.21$ μm (N = 58), which agrees well with our SEM results.
**Table 1: Characteristics of various airborne and engineered particles**

|  | Uranium oxide | Urban dust | Arizona dust | Hydrophilic $SiO_2$ | Hydrophobic $SiO_2$ | Graphene oxide |
|---|---|---|---|---|---|---|
| **Chemical composition (Minor components <2%)** | $UO_{2.191}$ | C, O, Si, S (Fe, Ca, Na, Al) | O, C, Si, (Fe, Ca, Na) | O, Si, N, C | O, Si, C | C, O, (S, Na) |
| **Crystalline** | $UO_2/U_3O_8$ | $SiO_2$ (quartz/ Amorphous) | $SIO_2$ (quartz) | $SiO_2$ (quartz/ Amorphous) | $SiO_2$ (quartz/ Amorphous) | Amorphous |
| **Density ($g/cm^3$)** | 10.65 | 2.1 (1.5~3.0) | 2.5 | 0.05* | 0.14* | 0.01~0.02* |
| **Surface area ($m^2/g$)** | N/A | 0.47 | N/A | 175–225 | 135–185 | >400 |
| **Observed Particle size (μm)** | 0.3–1.5 | 0.1–5 | 0.1–5 | 0.01 | 0.01 | 5–40 |
| **Calculated particle size (μm)** | $0.48 \pm 0.21$ (N = 58) | $1.21 \pm 0.48$ (N = 48) | $0.55 \pm 0.24$ (N = 40) | $7.11 \pm 3.91$ (N = 38) | $6.44 \pm 2.99$ (N = 38) | $28.20 \pm 6.56$ (N = 35) |

*tapped density
Dusts are usually solid particles that range in size from 1 to 100 μm in diameter. Depending on where the dust originates, the size,
shape, and chemical composition of the dust particles can differ dramatically. Researchers generally accept that particles with an
aerodynamic diameter that is >50 μm do not usually remain airborne for very long with a terminal velocity >7 cm/s. Conversely,
the terminal velocity of a 1 μm particle is about 0.03 mm/s (WHO, 1999). The terminal velocities of the urban and Arizona dusts
measured in this study were $0.293 \pm 0.511$ (N = 48) and $0.033 \pm 0.026$ mm/s (N = 40), respectively. Based on the value reported
by Whitby et al. (1957), an average approximate density of 2.1 $g/cm^3$ with a potential range of 1.5 to 3.0 $g/cm^3$ was chosen for the
urban dust. This approximation is acceptable since XRD and Raman analysis (Supplementary Information) showed that the urban
dust particles were primarily composed of $SiO_2$ ($\rho = 2.65$ $g/cm^3$) with minor contributions from impurities. The density of fine
street dust (<65 μm) was determined to be 1.6–1.8 $g/cm^3$ (Zhao et al., 2009). By using the terminal velocity and density of urban
dust, the average size of the urban dust particles was calculated to be $1.21 \pm 0.48$ μm. As reported by the supplier, Arizona dust
has a density of 2.5 $g/cm^3$, which is to be expected since the dust is mostly quartz (i.e., $SiO_2$). As such, Arizona dust has an average
calculated size of $0.55 \pm 0.24$ μm. For hydrophilic and hydrophobic $SiO_2$ nanoparticles and graphene oxide powders, the tapped
densities were used to determine particle size and surface charge. Because of the nano-scale size of the $SiO_2$ particles (~10 nm),
airborne particles agglomerate into micro-sized particles through various inter-particle forces, including the van der Waals force,
electrostatic force, and capillary force from moisture content. The average terminal velocities of the hydrophilic and hydrophobic
$SiO_2$ were $0.101 \pm 0.127$ (N = 38) and $0.217 \pm 0.221$ mm/s (N = 38), respectively, which are both faster than Arizona dust.
Therefore, the average calculated sizes of the hydrophilic and hydrophobic $SiO_2$ particles were $7.11 \pm 3.91$ and $6.44 \pm 2.99$ μm,
respectively. Finally, the calculated particle size for the spherical graphene oxide particles was $28.20 \pm 6.56$ μm (N = 35). These
results agree with the size of particles observed during SEM analysis.





**Figure 5: Surface charge distribution for airborne (a) UO$_2$, (b) urban dust, (c) Arizona dust, (d) hydrophilic SiO$_2$, (e) hydrophobic SiO$_2$, and (f) graphene oxide particles.**

Combining the size and free-fall velocities determined previously with the rise velocities determined through levitation experiments, Eq. 6 was used to calculate the charge $q$ of the observed particles. For the UO$_2$ particles, the average charge was $2.43 \times 10^{-18}$ C, with a standard deviation (SD) = $3.28 \times 10^{-18}$ C (N = 58 particles observed), or approximately 15 e, with SD = 20 e. The average charge carried by the urban dust particles was just over twice that amount at $5.12 \times 10^{-18}$ C, with SD = $1.12 \times 10^{-18}$ C, or approximately 32 e, with SD = 7 e., though the dust particles were also approximately 2.5 times larger. Arizona dust had the





lowest average charge out of all the particles examined; it had an approximate value of $1.16 \times 10^{-18}$ C, with SD = $1.87 \times 10^{-18}$ C,
or approximately 7 e, with SD = 12 e. For the direct comparison of these particles in the range of 0.5–1µm, the average charge of
$UO_2$ particle was $4.26 \times 10^{-18}$ C, with a standard deviation (SD) = $4.45 \times 10^{-18}$ C (N = 22 particles observed), or approximately 26
e (SD = 28 e), while the urban dust particles and Arizona dust particles were $8.60 \times 10^{-19}$ C (N=18) with a SD = $7.52 \times 10^{-19}$ C, or
approximately 5 e (SD = 4 e) and $2.05 \times 10^{-18}$ C, with a SD = $2.06 \times 10^{-19}$ C (N = 16), or approximately 12 e (SD = 12 e). The
charge of $UO_2$ is significantly higher than other natural-produced dust particles in the same range of particle size. Hydrophilic and
hydrophobic $SiO_2$ particles carried an order of magnitude more charge than the previously mentioned particles. The hydrophilic
$SiO_2$ particles had calculated charges of $3.17 \times 10^{-17}$ C with SD = $7.59 \times 10^{-17}$ C, or approximately 198 e with SD = 474 e. The
hydrophobic $SiO_2$ particles had calculated charges of $2.72 \times 10^{-17}$ C with SD = $3.49 \times 10^{-17}$ C, or approximately 170 e with
SD = 218 e. Graphene oxide particles were the most heavily charged; they had an average charge of $1.18 \times 10^{-16}$ C with
SD = $9.03 \times 10^{-17}$ C, or approximately 737 e with SD = 564 e. Thus, generally speaking, the charge carried by a particle was
proportional to its size, with larger particles carrying a greater charge on average. The radioactive uranium particles were the sole
exception to this trend since they carried substantially more charge than the slightly larger Arizona dust.
Figure 5 shows the charge distribution for each of the airborne particles examined as a function of the particle size. The magnitude
of the charge carried by the particles generally increased with increasing particle size, and the uranium particles deviated from this
trend. The charge carried by the $UO_2$ particles was more densely distributed than the Arizona or urban dusts, achieving a similar
magnitude of charge over a much smaller particle size range. Because of the additional filtration steps taken in their preparation,
$UO_2$ particles have a more uniform size distribution than the Arizona and urban dust particles. For the engineered particles, the
charge distribution was not only broader as a consequence of the agglomeration, but it was also higher; the maximum extent of the
surface charges was an order of magnitude above that of the dust particles. Nevertheless, the smallest $SiO_2$ particles (<10 µm)
showed considerable charge overlap with the $UO_2$ particles (<1 µm) between $10^{-18}$ and $10^{-17}$ C. Graphene oxide particles were yet
more heavily charged; many particles fell between $10^{-16}$ and $10^{-15}$ C, though these particles are also distributed over a much broader
size range. Thus, although the $UO_2$ particles were not as heavily charged as many of the other particles examined in this study,
they did possess a significantly higher charge density.
Based on the size and charge on the observed particles, the surface charge density for each of the particle populations examined in
this study was determined and the results are shown in Figure 6. The surface charge densities reported here are reasonable since
they are similar to results found in the literature. For example, the Arizona dust particles (mainly quartz) in this study had an
average surface charge density of $1.12 \pm 1.16$ µC/m$^2$, which is similar to the simulated value of 1.4 µC/m$^2$ for quartz particles with
a diameter of 132 µm reported by Toth et al. (2020). Additionally, Waitukaitis et al. (2014) found that for quartz particles with a
diameter ranging from 251 to 326 µm, the surface charge density was between 0.77 and 2.34 µC/m$^2$. $UO_2$ particles had a surface
charge density of $3.1 \pm 1.16$ µC/m$^2$, which is the highest charge density for any of the six particle classes and is approximately 2.75
times higher than the next highest surface charge density. To our knowledge, the surface charge density of $UO_2$ has not been
directly compared with other dust particles within the same system. The most obvious explanation for this enhanced charging of
the $UO_2$ particles is the self-charge generated from radioactive decay, though we must first consider whether or not the relatively
slow decay of uranium isotopes can generate the observed charging in a timely manner.

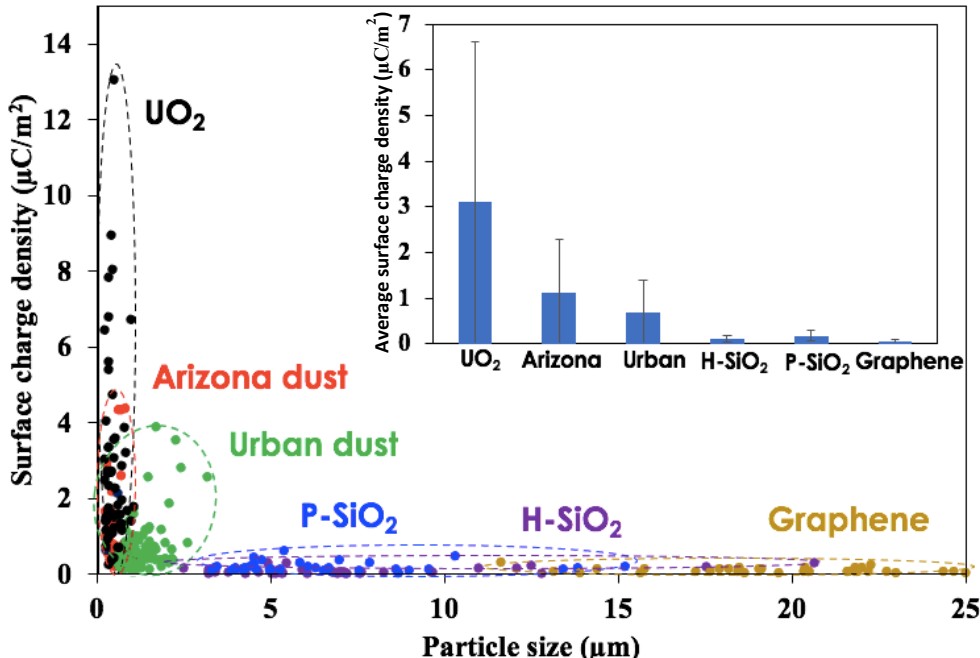

**Figure 6: Surface charge density distribution as a function of particle size. Inset is the average surface charge density for each of the**
**airborne particles examined in this study. Hydrophobic and hydrophilic SiO₂ are assigned to P-SiO₂ and H-SiO₂, respectively**

**3.2 Self-charging simulation results**
Self-charging from radioactive decay for various $UO_2$ particles was predicted using the methodology described in Section 2.4.
Decay simulations performed using IBIS assumed that $^{238}U$, $^{235}U$, and $^{234}U$ were the only radionuclides initially present in the $UO_2$
particle. At the start of the simulation, $^{238}U$, $^{235}U$, and $^{234}U$ accounted for 99.775, 0.22, and 0.005 mol % of all U in the system,
respectively. The decay chains for $^{235}U$ and $^{238}U$, which include the decay chain for $^{234}U$, are provided in the Supplementary
Information. Results from our charging simulations for $UO_2$ particles with a diameter of 0.2, 0.5, 0.8, and 1.0 μm over a period of
730 days when $M_\alpha$ was 6, 8, 10, and 12 are given in Figure 7. These results do not describe the charge of any individual particle
and should instead be interpreted as the average charge of each particle within a population of particles that are all the same
diameter. Thus, for example, the average charge of 0.436 e attained for particles with a diameter of 0.2 μm after 730 days when
$M_\alpha$ is 6 indicates that, across a population of 1,000 0.2 μm particles, the total charge carried by the population will be 436 e. Of
the isotopes considered in these simulations, $^{238}U$ and $^{234}U$ had by far the largest impact on particle self-charging, with the nuclides
together accounting for ~42.5% and ~45.8% of the charging rate when $M_\alpha$ was 6 and 12, respectively. Then, $^{234}Th$ and $^{234}Pa$ were
the next most substantial radioisotopes with the activity from each of those nuclides accounting for about 7.1% and 3.8% of the
predicted charge gained when $M_\alpha$ was 6 and 12, respectively. Finally, $^{235}U$ and $^{231}Th$ together accounted for just under 0.7% of the
predicted charge gained across all simulations. All other radioisotopes had a negligible contribution to the predicted particle charge
over the simulation run time.

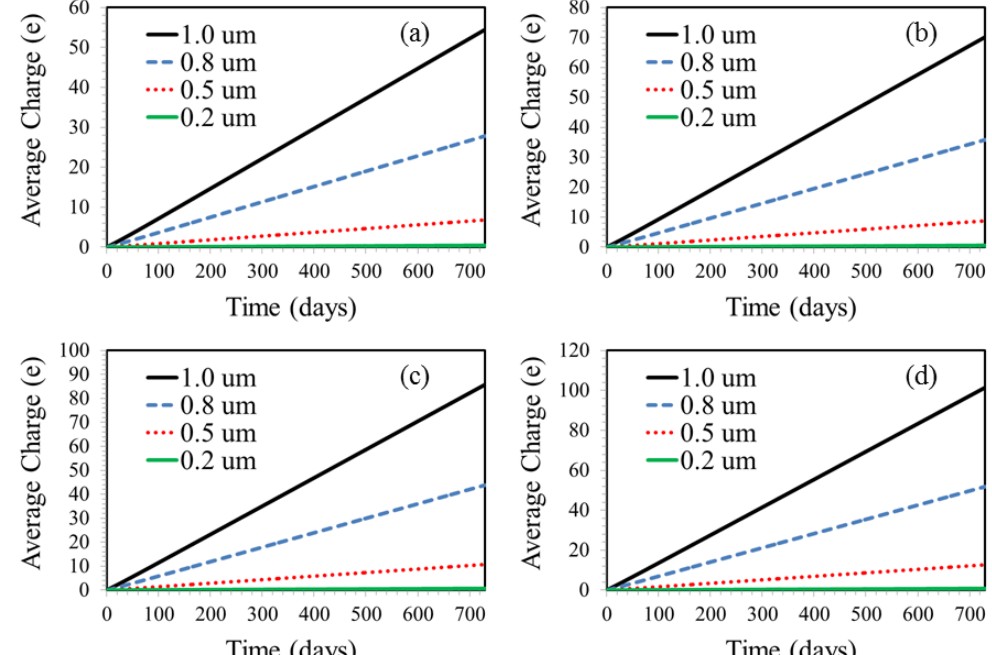

**Figure 7: Predicted average charge for UO$_2$ particles with diameters of 0.2, 0.5, 0.8, and 1.0 μm for a period of up to 730 days based on IBIS simulations of radioactive decay when the number of charges gained from each incident of alpha decay is (a) 6, (b) 8, (c) 10, and (d) 12.**

For particles with a diameter of 0.5 μm, the predicted average charge gained from radioactivity could be as low as 6.81 e when $M_\alpha$ was 6 or as much as 12.68 e when $M_\alpha$ was 12 after 730 days. Predicted particle self-charging increased proportionally with particle volume such that particles with a diameter of 0.8 μm had a charge between 27.84 and 51.84 e, and such that 1.0 μm particles attained a charge between 54.48 and 101.45 e when $6 \leq M_\alpha \leq 12$ over the same period. The experimental findings discussed previously showed that UO$_2$ particles with an approximate average particle size of 0.47 ± 0.21 μm had a surface charge that was roughly 8 e higher than particles of Arizona dust with an average size of 0.55 ± 0.24 μm. If we consider that the self-charging rate is directly proportional to the particle volume, then we can predict that the mean time required for a 0.47 μm particle to obtain a charge of 8 e purely through radioactive self-charging would be between 554.5 ($M_\alpha$ = 12) and 1,032.5 ($M_\alpha$ = 6) days. A larger 0.68 μm particle would take significantly less time, between 340.9 ($M_\alpha$ = 6) and 183.1 ($M_\alpha$ = 12) days, to achieve the same predicted self-charge. These results clearly indicate that a significant accumulation of charge is possible over an extended period of time for the UO$_2$ particles examined in this study. Nevertheless, given that the predicted rate of charging is relatively slow (0.14 e/day at most for 1.0 μm particles) the UO$_2$ particles were unlikely to develop significant charge during the experiment itself. Instead, the particles likely accumulated charge while in storage and already possessed a charge when they were initially prepared. This pre-experimental charging was, therefore, the likely reason for the higher surface charge density observed on the UO$_2$ particles. Also, Figure 6 shows significant variation in measured charge density, which may be attributed to the size range and the probabilistic nature of UO$_2$ decay.

## 4. Conclusions

An electrodynamic balance apparatus was employed to measure the surface charge of various airborne dust particulates, including particles of radioactive uranium oxide, Arizona dust, urban dust, hydrophilic silica, hydrophobic silica, and graphene oxide. The





surface charges and charge densities obtained were compared to understand the impact of radioactive self-charging. The average charge of uranium oxide was higher than expected given the size of the uranium particles and the highest surface charge density of any of the particles examined. Self-charge from radioactive decay of uranium was believed to be the origin of these higher charging characteristics. To assess the potential impact of self-charging, a charge-balance model was employed to predict charge accumulation on uranium particles of various sizes. Results from our model indicate that the higher-than-expected surface charge and charge density of the uranium oxide particles are likely associated with self-charging, though not within the experimental time frame.

**Code/Data availability.** Implicit Branching Isotope System **(**IBIS) code is available in Ecosystem Software – A C/C++ Library for Environmental Chemistry and Adsorption, https://bitbucket.org/gitecosystem/ecosystem

**Author contributions.** KM, CS and SY designed this research and acquired the financial support. GJ and AW performed the data analysis and wrote the manuscript.TS conducted XRD, Raman and IR analysis. JM prepared the sized $U_2O$ particle. AL provided the simulation tools and models used in the radioactive decay analysis. CS and SY supervised this project and reviewed the manuscript

**Competing interests.** The authors declare that they have no conflict of interest.

**Acknowledgments**

This work was supported by the Defense Threat Reduction Agency under grant number DTRA1-08-10-BRCWMD-BAA. The manuscript has been co-authored by UT-Battelle, LLC, under Contract No. DEAC05-00OR22725 with the US Department of Energy. The research was conducted at Oak Ridge National Laboratory (ORNL), which is managed by UT Battelle, LLC, for the US Department of Energy (DOE) under contract DE-AC05-00OR22725. Some of the materials characterization (SEM and X-ray diffraction) was conducted at the Center for Nanophase Materials Sciences (project ID: CNMS 2018-300), which is sponsored at ORNL by the Scientific User Facilities Division, US Department of Energy. The authors are grateful to Ms. Olivia Shafer for editing the manuscript.

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
