# Peer review of "Surface charge of environmental and radioactive airborne particles"

_Atmospheric Chemistry and Physics, 2021_

## Author Comment (AC1)

**OAK RIDGE NATIONAL LABORATORY**

MANAGED BY UT-BATTELLE FOR THE DEPARTMENT OF ENERGY

P.O. Box 2008
Oak Ridge, TN 37831-6181
Phone: (865) 241-3246
Fax: (865) 241-4829
tsourisc@ornl.gov

October 4, 2021

Dr. Ulrich Pöschl, Chief-executive Editor
Atmospheric Chemistry and Physics
Max Planck Institute for Chemistry, 55128 Mainz, Germany

Title: Surface charge of environmental and radioactive airborne particles
Author(s): Gyoung Gug Jang et al.
MS No.: acp-2021-417
MS type: Research article

Dear Dr. Ulrich Pöschl

Thank you for considering our manuscript for publication and for sending us the reviews. We appreciate the Reviewers' constructive comments. The manuscript has been revised to address all comments and suggestions from the Reviewers. The Reviewers' comments are copied verbatim below, followed by our responses and the corresponding revisions made to the manuscript. Each revision is highlighted and linked to a specific response. We hope that you and the reviewers will find our responses adequate and the revised manuscript acceptable for publication in the Atmospheric Chemistry and Physics.

Thank you for your valuable time and kind consideration.

Sincerely,

Costas Tsouris, Ph.D.
Oak Ridge National Laboratory
P.O. Box 2008
Oak Ridge, TN 37831-6181
E-mail: tsourisc@ornl.gov

**Reviewers' comments:**

**Reviewer#1**

The fate of airborne particles may be influenced by the charge, these particles carry. Charge in this context means on the one hand the net charge an ensemble of particles carries and on the other hand the (positive or negative) charge, the individual particles carry. In this manuscript the charging state of several types of particles, including radioactive uranium isotopes is investigated using an electrodynamic balance. This is an important topic; however, I have several concerns with the approach, presented here.

1) The charge, airborne particles carry after being in the atmosphere for some time, is mainly influenced by attachment of ions of both polarities. After long enough time a Boltzmann equilibrium is established (the time depends on the ion concentration). In this manuscript the charge of freshly suspended particles is investigated, this is the charge they achieve by the dispersion process. Any later changes are not even mentioned. So, these results do not reflect the charge of airborne particles.

> **Response:** The initial charge carried by individual particles such as $UO_2$, Urban dust, and Arizona dust has been determined through single particle levitation. Then, charge and ion balance equations, including self-charging and diffusion charging, have been employed to the charge evolution of the particles. In the revision, we included new experimental results, modeling, and discussion related to the triboelectric charging, self-charging, and diffusion charge of the airborne particles.

> **Revision:** The following text and figure were added on page 14 of the revised manuscript.

> "Charge evolution of triboelectrically charged $UO_2$ particles, urban dust, and Arizona dust in the atmosphere was predicted. Dust particles were chosen because they can be found in the atmosphere and may be triboelectrically charged during atmospheric transport. $UO_2$ particles were chosen to represent natural radioactive particles suspended in the atmosphere. The initial charge distributions of the particles and their mean size were obtained from measurements. Figure 8 shows time-dependent changes in the charge distribution of triboelectrically charged $UO_2$ particles, urban dust, and Arizona dust in air. As time elapsed, the particles lost their charge by capturing counterions. Although their initial charge distributions were dissimilar, all particles reached similar steady-state charge distributions within about 20 minutes. The timescale to reach a steady-state charge distribution can change when diffusion charging is enhanced or suppressed. When diffusion charging was enhanced by using high ion mobilities (e.g., $1.7 \, m^2 \, V^{-1} \, s^{-1}$ that was used by Harrison et al., 2010, to simulate the discharging of the Eyjafjallajökull volcanic ash plume), the charge distribution reached steady state within a shorter period of time. When diffusion charging was suppressed by using a lower ion production rate, e.g., only cosmic ray ionization of 4 ion pairs per second (Usoskin and Kovaltsov, 2006) and a higher particle concentration, e.g., $10^5 \, m^{-3}$ that can arise under urban atmospheric conditions (Kim et al., 2016), steady-state was still reached within 20 minutes, suggesting that triboelectrically charged particles can be quickly discharged by diffusion charging in the atmosphere."

[Figure]

**Figure 8. Charge evolution of $UO_2$ particles (a), urban dust (b), and Arizona dust (c) in air.**

2) For the uranium particles the charge, acquired directly by the decay process is discussed. However, the emitted radiation (α or β) will ionize the surrounding gas. The bipolar ions created thereby will attach to the particles and determine the particle charge. As one emitted a or b can produce a high number of ions this process is much more efficient than the direct charging, where (as shown in the manuscript) days are needed to achieve a high charging level. On average, this process will lead to a net charge close to zero, meaning that the concentration of negative and positive charges is equal this case there is no net driving force, acting on the particles. Should the charge distribution be unsymmetric, i.e., a net charge exists, this leads to a space charge which acts as driving force. Therefore, it would be important, to investigate both polarities in the experiment. In the manuscript no information

of polarity is given. These aspects have to be considered to get relevant information on the importance of particle charge.

**Response:** As the Reviewer pointed out, it is important to obtain the bipolar charge distribution of the particles because the distribution can be used to gauge the effects of particle charge on microphysical processes. Ionizing radiation can cause and enhance diffusion charging that can change the particle charge distribution to the Boltzmann distribution at steady state. Because of the low activity of the particles, ionizing radiation might not occur during the experiments. This has been clearly pointed out in the revised manuscript, on page 12, paragraph 3.

Because ionizing radiation can take place in the atmosphere and alter the particle charge distribution, the effects of diffusion charging on the charge distribution of triboelectrically charged particles were assessed using charge and ion balance equations. The initial triboelectrical charge of airborne particles such a $UO_2$, urban dust, and Arizona dust has been measured and the measurements were used for temporal prediction.

**Revision:** The following text and figure with references were added in Section 3.2 and Section 3.3.2 of the revised manuscript.

**"3.2 Self-charging simulation of single uranium oxide particles**
Since the experimental time scale might be too short to capture $UO_2$ self-charging, simulations were performed to assess the effects of radioactive decay on the charging of single radioactive particles. Self-charging from radioactive decay for single $UO_2$ particles was predicted using the methodology described in Section 2.4. Decay simulations performed using IBIS assumed that $^{238}U$, $^{235}U$, and $^{234}U$ were the only radionuclides initially present in single $UO_2$ particles and their mole fractions were 99.775, 0.22, and 0.005%, respectively. The decay chains for $^{235}U$, $^{238}U$, and $^{234}U$ are provided in the Supplementary Information.

Figure 7 shows simulation results for self-charging of $UO_2$ particles with a diameter of 0.2, 0.5, 0.8, and 1.0 µm and $M_\alpha = 6$. The initial activity for each particle size was $1.01\times10^{-9}$ Bq, $1.57\times10^{-8}$ Bq, $6.45\times10^{-8}$ Bq, and $1.25\times10^{-7}$ Bq, respectively. Because of the very low radioactivity levels, The particles slowly accumulated positive charge with time due to the decay of the uranium isotopes and their daughter radionuclides such as $^{234}Th$ and $^{234}Pa$. Larger $UO_2$ particles gained more positive charges because the particles contained more radioactive atoms. Of the radionuclides considered in these simulations, $^{238}U$ and $^{234}U$ had the largest impact on particle self-charging, with the radionuclides together accounting for ~42.5% of the charging rate. $^{234}Th$ and $^{234}Pa$ were the next most substantial radioisotopes with the activity from each of those radionuclides accounting for about 7.1% of the predicted charge. All other radionuclides had a negligible contribution to the charging of the $UO_2$ particles. These results demonstrate the importance of each radioisotope's activity (i.e. decay frequency) on the self-charging of radioactive particles. For example, the number concentration of $^{238}U$ is approximately four orders of magnitude greater than $^{234}U$ but both iosotopes are comparable with respect to the self-charging rate. Similarly, $^{234}Th$ and $^{234}Pa$ have concentrations that are more than seven orders of magnitude smaller than $^{234}U$ but, nevertheless, remain relevant to self-charging. For $M_\alpha = 8$, 10, and 12, the $UO_2$ self-charging effect was enhanced since each instance of alpha decay added more positive charges to the particles. These results suggest that the uranium oxide particles can spontaneously acquire positive charge via self-charging, but also that the self-charging rate is quite slow as a consequence of the decay rate of uranium.

[Figure]

**Figure 7: Predicted average charge for $UO_2$ particles with diameters of 0.2, 0.5, 0.8, and 1.0 µm for a period of up to 730 days based on IBIS simulations of radioactive decay when $M_\alpha = 6$.**

**"3.3.2. Particle charging and discharging by radionuclides and atmospheric ions**

The charging and discharging of particles in the atmosphere was predicted. We assumed that the radioactive particles undergo beta decay producing 2,067 ion pairs per decay and the activity of single radioactive particles is 1.4 Bq. These values refer to the ionization rate coefficient and the average activity of single $^{137}$Cs particles in the first radioactive plume of the Fukushima nuclear plant accident (Adachi et al., 2013; Kim et al., 2015). The measurements for the triboelectrically charged $UO_2$ particles were used as the initial charge distribution and the mean particle size for the prediction.

As time elapsed, the radioactive particles quickly lost their electrical charge because counterions were diffused onto the particles. However, while diffusion charging took place, the particles acquired positive charge by self-charging. For $A = 1.4$ Bq per particle, beta decay added one positive charge to each radioactive particle every 0.71 second. Because diffusion charging results in bipolar charging and self-charging leads to unipolar charging, these charging processes competed one another until reaching steady state. Figure 9 shows the steady-state charge distribution of the radioactive particles. Due to self-charging by beta decay, the steady-state charge distribution was shifted toward the right side compared to those seen in Figure 8a. In other words, the initially triboelectrically charged radioactive particles became unipolarly charged owing to self-charging. At steady-state, the major role of diffusion charging was to electrically neutralize net positive charge gained via self-charging. Similar results were obtained when the measurements for urban dust and Arizona dust and $A = 1.4$ Bq per particle were used as initial conditions for simulation. These results suggest that radioactivity can induce the charging of radioactive particles in the atmosphere and, if triboelectric charging does not continuously take place in radioactive plumes, the steady-state charge distribution of radioactive particles would depend on the competition of self-charging and diffusion charging."

[Figure]

**Figure 9. Steady-state charge distribution of initially triboelectrically charged radioactive particles in the atmosphere.**

Minor remarks:

3) The particle used are at least in part agglomerates and have shapes far from being spherical. Determining the surface/volume by assuming spherical particles therefore is problematic, this should at least be mentioned.

**Response**: The slip correction factor and shape factor of each particle have been considered to estimate the particle size with measurement and the charge and surface charge density have been recalculated accordingly in the revised manuscript.

**Revision:** The following text and equations were revised on pages 6 and 7 of the revised manuscript.

"The electric charge ($q$) carried by a particle can be determined using the formula:

$$q = \frac{m_p g (v_f + v_r)}{E v_f} \tag{9}$$

where

$$m_p = \frac{\pi d_p^3 \rho}{6} \tag{10}$$

$g$ is gravitational acceleration (m/s$^2$), $v_f$ is the velocity of the particle in free fall (m/s), $v_r$ is the upward velocity of the particle in a known electric field, $E$ is the electrtic field (V), $d_p$ is the equivalent volume diameter of

the particle (m) and $\rho$ is the particle density (kg/m³). A more detailed description of Eq. 10 is available in the Supplementary Information. The equivalent volume diameter of single particles can be given as (Hinds, 1999):

$$d_p = \sqrt{\frac{18 V_s \mu \chi}{\rho g C_c}} \qquad (11)$$

where $V_s$ is the terminal settling velocity of single particles in the particle levitator, $\mu$ is the air viscosity, $\chi$ is the shape factor, $\lambda_{mfp}$ is the mean free path, and $C_c$ is the Cunnungham correction factor determined using Eq. 12:

$$C_c = 1 + \frac{\lambda_{mfp}}{d_p}\left[1.257 + 0.4\exp\left(-\frac{1.1 d_p}{\lambda_{mfp}}\right)\right] \qquad (12)$$

4) How is the tapped density for SiO₂ particles determined? (table1)

**Response:** The material information was supplied by the vendor and was added in the Materials Section of the revised manuscript.

**Revision:** The following paragraphs were revised on pages 2 and 8 of the revised manuscript.

"High–surface area graphene oxide (N002-PDE: C = 60–80 atomic % and O = 10–30 atomic %, tapped density = 0.01–0.02 g/cm³) was purchased from Angstron Materials. The specific surfac area was measured at 497 m²/g (Jang et al. 2020). Hydrophobic (Aerosil R8200, tapped density = 0.14 g/cm³) and hydrophilic (Aerosil 200, tapped density = 0.05 g/cm³) fumed silica (SiO₂) nanoparticles with specific surface areas of 135–185 m²/g and 175–225 m²/g, respectively, were procured from Evonik Industries. Arizona dust with a size distribution of 0.97–352 μm (ISO 12103-1, A4 Coarse Grade) was obtained from Powder Technology, Incorporated. The specific surface area was measured at 0.31–0.34 m²/g. Additionally, NIST SRM 1649b urban dust was acquired from Sigma-Aldrich. In general, the specific surface area of urbran dust was 0.70 – 0.96 m²/g (Wang et. al., 2010).

We also added an explanation of the tapped density, which is shown as part of our response/revision to comment 5 by the Reviewer.

5) The great discrepancy between calculated and observed size of the SiO₂ particles (table1) is probably because observed are primary particles and calculated agglomerates, this should be discussed.

**Response:** The observed particle size is for a single particle and was determined by microscopic characterization and BET analysis, while the calculated particle size is determined using a formula based on the free falling velocity. UO₂ paraticles, urban dust, and Arizona dust show agreement in the size determination when bulk densities were applied. However, SiO₂ nanoparaticles and GO showed a discrepancy. Thus, we can interpret the calculated size as the size of agglomerates. In this case, instead of using a bulk density, a tapped density of these nanomaterials is more reasonable for the estimation of the agglomerated particle size. For example, when the bulk density of SiO₂ (2.65 g/cm³) is applied in the formula for a single particle, the average size of hydrophilic SiO₂ nanoparticle is determined to be 1.82 μm. This diameter is much higher than the observed single particle diameter of 0.01 μm. The slip correction factor and shape factor have been considered and the particle size was recalculated in the revised manuscript.

**Revision:** Table 1 has been revised on page 8 of the revised manuscript:

**Table 1: Characteristics of various airborne and engineered particles**

| | Uranium oxide | Urban dust | Arizona dust | Hydrophilic $SiO_2$ | Hydrophobic $SiO_2$ | Graphene oxide |
|---|---|---|---|---|---|---|
| **Chemical composition (Minor components <2%)** | $UO_{2.191}$ | C, O, Si, S (Fe, Ca, Na, Al) | O, C, Si, (Fe, Ca, Na) | O, Si, N, C | O, Si, C | C, O, (S, Na) |
| **Crystalline** | $UO_2/U_3O_8$ | $SiO_2$ (quartz/ Amorphous) | $SiO_2$ (quartz) | $SiO_2$ (quartz/ Amorphous) | $SiO_2$ (quartz/ Amorphous) | Amorphous |
| **Density ($g/cm^3$)** | 10.65 | 2.1 (1.5~3.0) | 2.5 | 0.05* | 0.14* | 0.01~0.02* |
| **Surface area ($m^2/g$)** | 0.4–1.8# | 0.70–0.96 | 0.31–0.34 | 175–225 | 135–185 | 497 |
| **Observed single particle size (μm)** | 0.3–1.5 | 0.1–5 | 0.1–5 | 0.01 | 0.01 | 5–40 |
| **Calculated particle size (μm)** | $0.65 \pm 0.28$ (N = 180) | $1.47 \pm 0.53$ (N = 88) | $0.90 \pm 0.52$ (N = 88) | $8.37 \pm 4.59$ (N = 38) | $7.62 \pm 3.49$ (N = 38) | $23.82 \pm 5.54$ (N = 33) |

*tapped density, # calculated based on a formula: Diameter (nm) = 6000/[(BET surface area in $m^2/g$) ×(density in $g/cm^2$)]

Also, the following text was added on page 8 of the revised manuscript.

"For hydrophilic and hydrophobic $SiO_2$ particles and graphene oxide powders, the tapped densities were used to determine particle size and surface charge. Tapped density of a powder is the ratio of the mass of the powder to the volume occupied by the powder after it is tapped for a certain period of time. In general, the density of nanoparticle is size-dependent. Because of the nano-scale size of individual $SiO_2$ particles (~10 nm), airborne particles agglomerate into micron-diameter particles through various inter-particle forces, including the van der Waals attractive force. The average terminal velocities of the hydrophilic and hydrophobic $SiO_2$ were $0.101 \pm 0.127$ (N = 38) and $0.217 \pm 0.221$ mm/s (N = 38), respectively, which are both greater than that of Arizona dust. Therefore, the average calculated sizes of the hydrophilic and hydrophobic $SiO_2$ particles were $8.37 \pm 4.59$ and $7.62 \pm 3.49$ μm, respectively. Finally, the calculated particle size for the spherical graphene oxide particles was $23.82 \pm 5.54$ μm (N = 33). These results agree with the size of particles observed during SEM analysis."

**Reviewer #2**
The charge of different aerosol particles is determined by means of an electrodynamic balance using optical microscopy for particle monitoring. From the comparison of sink velocities in free fall and migration velocities in an applied electric field, statements about density, size and charge of the particles are derived. Besides mineral dust powders and agglomerated engineered particles, radioactive uranium oxide particles are also investigated, for which self-charging is suspected as a result of radioactive beta and alpha decay. While the manuscript is sound in the parts related to radioactivity aspects and the objectives become clear, it contains many inaccuracies and the conclusions made are not supported by the data presented. In particular, key aspects of particle characterization and charging remain unexplained or are not even addressed. Therefore, the manuscript cannot be recommended for publication in the present form.

Particular comments:
1) Page 3: The determination of the electrical charge is explained using the Milikan setup in Fig.2. This is fine for dense spherical objects like oil droplets. But the particles used sometimes deviate considerably from dense and spherical morphologies. Here, effective densities and especially shape factors can be used, but this is later only marginally taken up for densities (tagged density for GO agglomerates) and not at all with respect to shape (e.g., in the manner of sphericity or dynamic shape factors).

> **Response**: The slip correction factor, shape factor, and effective density of each particle have been considered to estimate the equivalent volume diameter of the particles and the charge and surface charge density have been recalculated accordingly.

> **Revision:** The following text and equations were revised on pages 6 and 7 of the revised manuscript.

"The electric charge ($q$) carried by a particle can be determined using the formula:

$$q = \frac{m_p g (v_f + v_r)}{E v_f} \tag{9}$$

where

$$m_p = \frac{\pi d_p^3 \rho}{6} \tag{10}$$

$g$ is gravitational acceleration (m/s$^2$), $v_f$ is the velocity of the particle in free fall (m/s), $v_r$ is the upward velocity of the particle in a known electric field, $E$ is the electrtic field (V), $d_p$ is the equivalent volume diameter of the particle (m) and $\rho$ is the particle density (kg/m$^3$). A more detailed description of Eq. 10 is available in the Supplementary Information. The equivalent volume diameter of single particles can be given as (Hinds, 1999):

$$d_p = \sqrt{\frac{18 V_s \mu \chi}{\rho g C_c}} \tag{11}$$

where $V_s$ is the terminal settling velocity of single particles in the particle levitator, $\mu$ is the air viscosity, $\chi$ is the shape factor, $\lambda_{mfp}$ is the mean free path, and $C_c$ is the Cunnungham correction factor determined using Eq. 12:

$$C_c = 1 + \frac{\lambda_{mfp}}{d_p} \left[ 1.257 + 0.4 \exp\left( -\frac{1.1 d_p}{\lambda_{mfp}} \right) \right] \tag{12}$$

2) P 4: The abbreviations "ICDD", "GSASII" and "CIF files" are not explained anywhere.

   **Response:** An explanation of these accronyms has been provided in the revised manuscript.

   **Revision:** The following revised text has been added on page 4 of the revised manuscript.

   Samples of uranium, urban, and desert dust particles were identified using the International Centre for Diffraction Data (ICDD) powder diffraction file (PDF4+). General Structure Analysis System (GSASII) was used to determine $U_3O_8$ and $UO_2$ phase fractions. Crystallographic Information File (CIF) data for the two phases were imported, and a light refinement of the lattice parameters was performed.

3) P 5, middle: "Before blowing the particles into the balance chamber, the pipette was vigorously shaken for 1 min to induce charging though the triboelectric effect." This is one of the main points of criticism. Triboelectric charging of particles is an extremely complex process that depends not only on the material (e.g. work functions of particle and walls), but also on the type and number of collisions, on adsorbates and on environmental conditions. Good review articles on this can be found, for example, in Matsusaka et al. (2010), Mirkowska et al. (2016), Zou et al. (2019) and Lacks & Shinbrot (2019). Even for particles of the same size from the same material, bipolar charge distributions are observed and the charge polarity can even switch from negative to positive with increasing number of collisions. Therefore, the type of particle charging presented here is completely undefined and is not suitable for drawing conclusions about the charging behavior of different particles (varying in material, shape, internal structure, surface roughness, surface conductivity, etc.).

   **Response:** The effect of triboelectric charging on $UO_2$ particles in this system has been quantified using additional experiments. Also, the charge polarity of other airborne dust particles was measured. Other factors such as dynamic shape factor and other correction factors have been considered and the charge and surface charge density distribution have been recalculated as a function of particle size. As the reviewer pointed out, triboelectric charging is affected by various parameters. In order to define the origin of electrical charge of the particles (i.e., triboelectric charging vs self-charging), an additional experiment was performed with particles falling without shaking. The experiment demonstrated that the $UO_2$ particles could be charged by triboelectric charging and that the observation of $UO_2$ self-charging might be limited during the experimental time-scale because uranium isotopes slowly undergo radioactive decay. On the basis of this experiment, we revised the manuscript to discuss the origin of the particle charging in the levitator and the effects of self-charging on the charging of radioactive particles.

To investigate the charging behavior of different particles in the atmosphere, new simulations have been performed based on charge and ion balance equations, and text has been added to discuss simulation results. The simulation results indicated that, in open air, triboelectrically charged nonradioactive particles can be quickly discharged by capturing counterions, but triboelectrically charged radioactive particles can remain strongly charged due to radioactivity-induced self-charging. The manuscript has been revised based on these results.

**Revision:**
"**Abstract.** The charging of various airborne particles was investigated using single particle levitation and charge balance equations. Though radioactive decay and triboelectrification can induce charging, it is typically assumed that the aerosols in a radioactive plume will not carry significant charge at steady state since atmospheric particles can have their charge neutralized through the capture of naturally occurring counterions (i.e., diffusion charging). To assess this assumption, we began by measuring the charge and surface charge density of various triboelectrically charged radioactive and non-radioactive particles. These measurements were then used as a basis for the prediction of particle charge evolution in open air. By using electric field–assisted particle levitation in air, the charge, charge distribution, and surface charge density were determined for triboelectrically charged uranium oxide aerosols (<1 μm), urban dust, Arizona desert dust, hydrophilic and hydrophobic silica nanoparticles, and graphene oxide powders. Of these particles, uranium oxide aerosols exhibited the highest surface change density. Charge balance equations were employed to predict average charge gained from radioactive decay as a function of time and to evaluate the effects of diffusion charging on triboelectrically charged radioactive and nonradioactive particles in the atmosphere. Simulation results show that particles, initially charged through triboelectrification, can be quickly discharged by diffusion charging in the absence of radioactive decay. Nevertheless, simulation results also indicate that particles can be strongly charged when they carry radionuclides. These experimental and simulation results suggest that radioactive decay can induce strong particle charging that may potentially affect atmospsheric transport of airborne radionuclides."

Page 9-10: Figure 5 shows the triboelectrification induced charge distribution for each of the airborne particles examined as a function of the particle size. Combining the size and free-fall velocities determined previously with the rise velocities determined through levitation experiments, Eq. 9 was used to calculate the charge q of the single particles observed. Considering the events responsible for airborne particle generation, we assume that the initial charge characterisitcs of the airborne particles are mostly a consequence of interparticle friction arising form particle lift and collisons during transportation. All uranium species initially contained in the $UO_2$ particles were radioactive, but the activity of the particles was very low because the uranium isotopes are long-lived radionuclides, indicating that self-charging of the particles may be limited during these measurements. Figures 5a and 5b show that the polarized charges measured after inducing triboelectrification through shaking were significantly higher than the charges measured without shaking the particles. Unshaken particles were measured by dropping particles into the chamber of the electrodynamic balance without blowing. Then, the initial charge of all other particles was measured. The initial charge was increased by the triboelectric charging effect via shaking.

Page 11: "To our knowledge, the surface charge density of $UO_2$ particles has not been directly compared with other dust particles within the same system. These results are not particularly supprising given the complex nature of triboelectric charging which depends on a number of different factors including the marterial being charged, the freguency at which particles collide, and the environment in which charging occurs (Matsusaka et al., 2010; Mirkowska et al., 2016; Zou et al., 2019). Thus, while much of the discrepancy between the charging behavior of $UO_2$ and the other particle classes can be attributed to the excentricities of triboelectric charging, we cannot totally discount the effect of self-charging at this stage. Additional examination of the potential for $UO_2$ self-chaging will be necessary to assess how much of an impact it has on the particle charge in the context of these experiments."

"**4. Conclusions**
An electrodynamic balance apparatus was employed to measure the charging of various airborne dust particulates, including particles of uranium oxide, Arizona dust, urban dust, hydrophilic silica, hydrophobic silica, and graphene oxide. The charge, charge distribution, and charge density data obtained were compared to understand the triboelectric charging characteristics of the particles. The measurements showed that uranium oxide can be more easily charged by triboelectrification than the other particles. Charge-balance equations were employed to simulate the charging of triboelectrically charged radioactive and nonradioactive

dust particles. Simulation results indicated that, in open air, triboelectrically charged nonradioactive particles can be quickly discharged by capturing counterions but radioactive particles can remain strongly charged because of radioactive self-charging. These results suggest that various electrostatic interactions involing radioactive particles can be created within dispersed radioactive plumes, and the effects of electrostatic interactions on dispersion of radioactive plumes should be further investigated."

[Figure]

**Figure 5: Triboelectrification induced surface charge distribution for airborne (a) UO₂ without (w/o) shaking (b) UO₂ with (w/) shaking, (c) Urban Dust w/ shaking, (d) Arizona Dust w/ shaking (e) Hydrophobic and Hydrophilic SiO₂ nanoparticles w/ shaking, and (f) graphene oxide particles w/ shaking. [-] and [+] indicate negative and positive charges, respectively.**

[Figure]

**Figure 6: a) Triboelectrically charged negative surface charge density distribution as a function of particle size. b) Triboelectrically charged postive surface charge density distribution. c) The average surface charge density for each of the airborne particles examined in this study. Hydrophilic and hydrophobic SiO₂ are assigned to HPH-SiO₂ and HP-SiO₂, respectively.**

4) P 5: "From among the particles that were in view, a number of particles were selected, and their velocities were monitored under various electric fields to determine their approximate size, density, and charge." If I understand the procedure correctly, sink velocity and migration velocity in the electric field were determined on individual particles using microscopy. Only if $v_f$ and $v_r$ were measured for each individual particle, the presented evaluation makes sense. Otherwise, one has only average values, which would not allow a representation of charge vs. size, as shown in Fig. 5. Do the light spots shown in Fig.3 really come from UO₂ particles with diameters of 0.5 μm?

    **Response:** The $v_f$ (sink velocity) and $v_r$ (migration velocity) of each individual particle were determined based on the applied potential. From these measurements, the size of each particle examined is determined independently. The light spots shown in Fig. 3 come from a UO₂ particle, but the diameter of the particle can be different.

5) P 6, Eq. (6): Since the pressure-dependent term in Eq.(6) is an approximation of the Cunningham correction, it would be helpful to explain for which Knudsen numbers this approximation is valid. In addition, as mentioned above, the shape influence of the particles was not taken into account in the derivation of this equation.

    **Response**: The shape factor of each particle and the Cunninham correction have been considered and the charge and surface charge density have been recalculated accordingly.

    **Revision:** The following text and equations were revised on pages 6 and 7 of the revised manuscript.

"The electric charge ($q$) carried by a particle can be determined using the formula:

$$q = \frac{m_p g (v_f + v_r)}{E v_f} \qquad (9)$$

where

$$m_p = \frac{\pi d_p^3 \rho}{6} \qquad (10)$$

$g$ is gravitational acceleration (m/s²), $v_f$ is the velocity of the particle in free fall (m/s), $v_r$ is the upward velocity of the particle in a known electric field, $E$ is the electrtic field (V), $d_p$ is the equivalent volume diameter of the particle (m) and $\rho$ is the particle density (kg/m³). A more detailed description of Eq. 10 is available in the Supplementary Information. The equivalent volume diameter of single particles can be given as (Hinds, 1999):

$$d_p = \sqrt{\frac{18 V_s \mu \chi}{\rho g C_c}} \tag{11}$$

where $V_s$ is the terminal settling velocity of single particles in the particle levitator, $\mu$ is the air viscosity, $\chi$ is the shape factor, $\lambda_{mfp}$ is the mean free path, and $C_c$ is the Cunnungham correction factor determined using Eq. 12:

$$C_c = 1 + \frac{\lambda_{mfp}}{d_p}\left[1.257 + 0.4\exp\left(-\frac{1.1 d_p}{\lambda_{mfp}}\right)\right] \tag{12}$$

6) P 7: The densities of $UO_2$ and $U_3O_8$ are reversed.

   **Response:** The densities have been fixed as recommended.

   **Revision:** Page 8:"Additionally, x-ray diffraction indicated that the $UO_2$ ($\rho = 10.97$ g/cm$^3$) and $U_3O_8$ ($\rho = 8.3$ g/cm$^3$) phase fractions were 0.8819 and 0.1181, respectively (see Supplementary Information).

7) P 7: Where does the "tapped density" of the GO particles come from?

   **Response**: The tapped density of the GO particles comes from the vendor. The information was added in the experimental section. The following text was revised on pages 2 and 8 of the revised manuscript.

   **Revision**:

   Page 2: "High–surface area graphene oxide (N002-PDE: C = 60–80 atomic % and O = 10–30 atomic %, tapped density = 0.01–0.02 g/m$^3$) was purchased from Angstron Materials. The specific surfac area was measureded to 497 m$^2$/g (Jang et al. 2020). Hydrophobic (Aerosil R8200, tapped density = 0.14 g/m$^3$) and hydrophilic (Aerosil 200, tapped density = 0.05 g/m$^3$) fumed silica ($SiO_2$) nanoparticles with specific surface areas of 135–185 and 175–225 m$^2$/g, respectively, were procured from Evonik Industries. Arizona dust with a size distribution of 0.97–352 µm (ISO 12103-1, A4 Coarse Grade) was obtained from Powder Technology, Incorporated. The specific surfac area was measureded to 0.31–0.34 m$^2$/g. Additionally, NIST SRM 1649b urban dust was acquired from Sigma-Aldrich. A specific surface area of urbran dust was determined to 0.70–0.96 m$^2$/g (Wang et. al., 2010)."

   Also, the following text was added on page 8 of the revised manuscript.

   "For hydrophilic and hydrophobic $SiO_2$ particles and graphene oxide powders, the tapped densities were used to determine particle size and surface charge. Tapped density of a powder is the ratio of the mass of the powder to the volume occupied by the powder after it is tapped for a certain period of time. In general, the density of nanoparticle is size-dependent. Because of the nano-scale size of individual $SiO_2$ particles (~10 nm), airborne particles agglomerate into micron-diameter particles through various inter-particle forces, including the van der Waals attractive force. The average terminal velocities of the hydrophilic and hydrophobic $SiO_2$ were $0.101 \pm 0.127$ (N = 38) and $0.217 \pm 0.221$ mm/s (N = 38), respectively, which are both greater than that of Arizona dust. Therefore, the average calculated sizes of the hydrophilic and hydrophobic $SiO_2$ particles were $8.37 \pm 4.59$ and $7.62 \pm 3.49$ µm, respectively. Finally, the calculated particle size for the spherical graphene oxide particles was $23.82 \pm 5.54$ µm (N = 33). These results agree with the size of particles observed during SEM analysis."

8) P 8: "Combining the size and free-fall velocities determined previously with the rise velocities determined through levitation experiments, Eq. 6 was used to calculate the charge q of the observed particles." Here it sounds as if the free-fall-sink velocities were measured first and then the migration velocities in the electric field were measured on other particles (of the same material). As explained above, this makes no sense at all and leads to unmeaningful results, as can be seen from the huge standard deviations (in part significantly larger than the mean value).

**Response:** We measured the free-fall-sink velocity of a single particle to estimate the particle size and then the upward migration velocities in the electric field for the same particle. Each particle has its own size and charge. Also, statistical analysis has been performed to understand the significance of the surface charge density distribution for each particle type. This analysis suggests that the $UO_2$ particles behave differently from other airborne particles. This is discussed in the revised manscript, and the results are available in the Supplementary Information.

**Revision**:

"To understand the significance of the measured charge densities, we performed analysis of variance (ANOVA) for testing the equality of means of charge density over different particles. The results indicate that not all of the means are equal for both positive and negative charge density. To identify significant pair-wise difference, Tukey's range test is conducted. As a result, two pairs, $UO_2$-Arizona dust and $UO_2$-Urban dust, show significant difference for positive density, while all pairs including $UO_2$ have significant difference for negative density. In addition, we confirm that the difference between without and with shaking for both positive and negative densities in $UO_2$ is significant. All statistical hypotheses are tested at significant level 0.05. The results are available in supplementary information."

[Figure]

**Figure S14.** Density estimation of charge density. The x-axis is the value of charge density, while the y-axis represents the estimated density. Top: Before vs. After shaking in $UO_2$, Middle: Positive vs. Negative in $UO_2$, Bottom: Positive vs. Negative in Arizona and Urban.

9) P 8/P 9: "The average charge carried by the urban dust particles was just over twice that amount at $5.12 \times 10^{-18}$ C, with SD = $1.12 \times 10^{-18}$ C, or approximately 32 e, with SD = 7 e, though the dust particles were also approximately 2.5 times larger." The comparison of uranium oxide with urban dust makes no sense. For one thing, only a vague average value of the diameter is 2.5 times larger, and for another, different materials charge triboelectrically in a completely different way. Therefore, apples and oranges are compared here. Also the significant charging difference between uranium oxide and Arizona dust is, except for one point, not recognizable. The statement that $UO_2$ is substantially more charged cannot be recognized in view of the enormous ranges of the standard deviations.

**Response:** Please see our response to the previous comment since the revisions made there also address this comment. As mentioned above, a statistical analysis has been performed to understand the meaning of the surface charge density distribution for each particle class. It suggests that the $UO_2$ particles are different from the other airborne particles examined. Additional data analysis and discussion were added on pages 11 and 12 to explain the new experimental data; however, the message has been significantly changed.

**Revision:** The average charge measured for each particle class is summarized in Table 2. Negatively charged urban dust particles had, on average, about 2.6 times more charge than negative $UO_2$ particles though the dust particles were also approximately 2.3 times larger. Arizona dust had the lowest average charge out of all the particles examined. For the direct comparison of these particles in the range of 0.5–1µm, the average negative charge of $UO_2$ particle was $-4.86 \times 10^{-18}$ C, with a standard deviation (SD) = $5.54 \times 10^{-18}$ C (N = 28 particles observed), or approximately 30 e (SD = 35 e), while the urban dust particles and Arizona dust particles were $-1.14 \times 10^{-18}$ C (N=16) with a SD = $1.12 \times 10^{-18}$ C, or approximately 7 e (SD = 7 e) and -$2.07 \times 10^{-18}$ C, with a SD = $2.83 \times 10^{-18}$ C (N = 20), or approximately 13 e (SD = 18 e), respectively. For positive charges, the average charge of $UO_2$ particle was $7.20 \times 10^{-18}$ C, with a standard deviation (SD) = $9.37 \times 10^{-18}$ C (N = 45 particles observed), or approximately 45 e (SD = 59 e), while the urban dust particles and Arizona dust particles were $1.08 \times 10^{-18}$ C (N=9) with a SD = $6.01 \times 10^{-19}$ C, or approximately 7 e (SD = 4 e) and $9.93 \times 10^{-19}$ C, with a SD = $5.82 \times 10^{-19}$ C (N = 26), or approximately 6 e (SD = 4 e), respecively. Thus, the charge carried by the $UO_2$ particles is, on average, significantly higher than that of other naturally-produced dust particles in the same size range. Hydrophilic and hydrophobic $SiO_2$ particles carried an order of magnitude more charge than the previously mentioned particles. Graphene oxide particles were the most heavily charged. Thus, in general, the charge carried by a particle was proportional to its size, with larger particles carrying a greater charge on average. The radioactive uranium particles were the sole exception to this trend since they carried substantially more charge than the slightly larger Arizona dust.

Based on the size and charge on the observed particles, the surface charge density for each of the particle populations examined in this study was determined and the results are shown in Figure 6. The surface charge densities reported here are reasonable since they are similar to results found in the literature. For example, the Arizona dust particles (mainly quartz) in this study had an average negative surface charge density of $1.13 \pm 1.19$ µC/m$^2$, which is similar to the simulated value of 1.4 µC/m$^2$ for quartz particles with a diameter of 132 µm reported by Toth et al. (2020). Additionally, Waitukaitis et al. (2014) found that for quartz particles with a diameter ranging from 251 to 326 µm, the surface charge density was between 0.77 and 2.34 µC/m$^2$. $UO_2$ particles had a negative surface charge density of $3.44 \pm 3.89$ µC/m$^2$, which is the highest charge density for any of the six particle classes and is approximately 3 times higher than the next highest surface charge density. To our knowledge, the surface charge density of $UO_2$ has not been directly compared with other dust particles within the same system. These results are not particularly supprising given the complex nature of triboelectric charging which depends on a number of different factors including the material being charged, the freguency at which particles collide, and the environment in which charging occurs (Matsusaka et al., 2010; Mirkowska et al., 2016; Zou et al., 2019). Thus, while much of the discrepancy between the charging behavior of $UO_2$ and the other particle classes can be attributed to the excentricities of triboelectric charging, we cannot totally discount the effect of self-charging at this stage. Additional examination of the potential for $UO_2$ self-chaging will be necessary to assess how much of an impact it has on the particle charge in the context of these experiments.

**Table 2: Average charge measurements for airborne and engineered particles**

| | Uranium Oxide | | Urban Dust | Arizona Dust | Hydrophilic SiO$_2$ | Hydrophobic SiO$_2$ | Graphene oxide |
|---|---|---|---|---|---|---|---|
| | W/O shaking | W/ shaking | W/ shaking | W/ shaking | W/ shaking | W/ shaking | W/ shaking |
| Negative charge | -7.13 × 10$^{-19}$ C ± 7.03 × 10$^{-19}$ C (n=30) | -3.27 × 10$^{-18}$ C ± 4.41 × 10$^{-18}$ C (n=60) | -8.56 × 10$^{-18}$ C ± 1.84 × 10$^{-17}$ C (n=48) | -1.53 × 10$^{-18}$ C ± 2.32 × 10$^{-18}$ C (n=48) | -5.12 × 10$^{-17}$ C ± 1.23 × 10$^{-16}$ C (n=38) | -4.28 × 10$^{-17}$ C ± 5.45 × 10$^{-17}$ C (n=38) | -1.49 × 10$^{-16}$ C ± 1.23 × 10$^{-16}$ C (n=33) |
| Positive charge | 3.53 × 10$^{-18}$ C ± 3.51 × 10$^{-18}$ C (n=3) | 7.88 × 10$^{-18}$ C ± 1.01 × 10$^{-17}$ C (n=60) | 8.82 × 10$^{-18}$ C ± 9.37 × 10$^{-18}$ C (n=40) | 2.95 × 10$^{-18}$ C ± 3.62 × 10$^{-18}$ C (n=40) | N/A | N/A | N/A |

10) P 9/P 10: "Based on the size and charge on the observed particles, the surface charge density for each of the particle populations examined in this study was determined and the results are shown in Figure 6." The derivation of Fig.6 contains some doubtful assumptions. Although in Table 1 the surface area of uranium oxide was not determined (N/A), a value must be assumed here to calculate the surface charge density. However, this value is not given. On the other hand, for SiO$_2$ and GO, the primary particle sizes are used to determine the specific surface area. This reduces the surface charge, although the structure of the agglomerated particles has not been characterized and therefore no information about the charge distribution over these particles is available. Especially hydrophobic SiO$_2$ has a very low surface conductivity, so that charges accumulate only on the outer surface and the surface specific charge is significantly higher. On the other hand, in Fig. 6, agglomerate diameters are used for the size of SiO$_2$ and GO particles. This is not correct.

**Response:**
The specific surface areas of SiO$_2$ was supplied by the vendor. The values of GO and Arizona dust were experimentally measured in previous work (Jang et al. Composites Science and Technology 199, 108352, 2020 and Jang et al, Nanoscale Advances 1 (3), 1249-1260, 2019), not determined by the primary particle size (e.g., SEM image analysis). No BET instruments are available to measure radioactive particles due to contamination issues.

The indivual particle size was determined by free-fall-sink velocity and the material density and then the surface charge was calculated. Even though we can estimate the particle size by the surface area, the size is an average value, not for a single particle. It cannot reflect the individual particle size obtained in this work.

Using a formula, i.e., Diameter (nm) = 6000/(BET surface area in m$^2$/g) x (density in g/cm$^3$) with size in nm (10$^{-9}$ m) and a density of SiO$_2$ (2.65 g/cm$^3$), the primary particle sizes of hydrophobic and hydrophilic SiO$_2$ nanoparticles were determined at 12.2−16.6 nm and 10.0−12.9 nm, respectively. These sizes agree well with the SEM observed particle size. However, when the bulk density of SiO$_2$, i.e., 2.65 g/cm$^3$, was used to determine the size for a single particle observed in the levitator, the average size of single SiO$_2$ nanoparticles is determined at ~1.82 $\mu$m. This diameter is much higher than the primary particle size, which is ~0.01 $\mu$m. Thus, it is reasonable to assume that the observed single partcle in the levitator is actually an agglomerate. The tapping density is closer to the density of the agglomerates, thus it can be used for determination of the surface charge density.

Also, graphene oxide used in this study is a single sheet or a few layers of atomic carbon sheets. In the atmosphere, GO sheets agglomerate into a nearly spherical particle to reduce their surface energy. We assumed that the diameter of the agglomerates, determined by the tapped density, is more suitable for the estimation of the particle size in the atmosphere. Based on this discussion, the surface charge of each particle was determined.

**Revision:**

**Table 1: Characteristics of various airborne and engineered particles**

| | Uranium oxide | Urban dust | Arizona dust | Hydrophilic $SiO_2$ | Hydrophobic $SiO_2$ | Graphene oxide |
|---|---|---|---|---|---|---|
| **Chemical composition (Minor components <2%)** | $UO_{2.191}$ | C, O, Si, S (Fe, Ca, Na, Al) | O, C, Si, (Fe, Ca, Na) | O, Si, N, C | O, Si, C | C, O, (S, Na) |
| **Crystalline** | $UO_2/U_3O_8$ | $SiO_2$ (quartz/ Amorphous) | $SIO_2$ (quartz) | $SiO_2$ (quartz/ Amorphous) | $SiO_2$ (quartz/ Amorphous) | Amorphous |
| **Density (g/cm$^3$)** | 10.65 | 2.1 (1.5~3.0) | 2.5 | 0.05* | 0.14* | 0.01~0.02* |
| **Surface area (m$^2$/g)** | 0.4–1.8# | 0.70–0.96 | 0.31–0.34 | 175–225 | 135–185 | 497 |
| **Observed single particle size (μm)** | 0.3–1.5 | 0.1–5 | 0.1–5 | 0.01 | 0.01 | 5–40 |
| **Calculated particle size (μm)** | 0.65 ± 0.28 (N = 180) | 1.47 ± 0.53 (N = 88) | 0.90 ± 0.52 (N = 88) | 8.37 ± 4.59 (N = 38) | 7.62 ± 3.49 (N = 38) | 23.82 ± 5.54 (N = 33) |

*tapped density, # calculated based on a formula: Diameter (nm) = 6000/[(BET surface area in m$^2$/g) ×(density in g/cm$^2$)]

[Figure]

**Figure 6: a) Triboelectrially charged negative surface charge density distribution as a function of particle size. b) Triboelectrially charged postive surface charge density distribution. c) The average surface charge density for each of the airborne particles examined in this study. Hydrophilic and hydrophobic SiO₂ are assigned to HPH-SiO₂ and HP-SiO₂, respectively.**

11) P 10/P 11: Self-charging simulation results: The discussion is very lengthy and ends with the uranium oxide particles unlikely to have developed a significant charge during the experiment. Therefore, it remains incomprehensible why in the conclusions it is said that "Self-charge from radioactive decay of uranium was believed to be the origin of these higher charging characteristics."

**Response:** The previous conclusion has changed in the revision, because the experimental time-scale might be too short to clearly observe a significant effect on UO₂ particle charge. Although all uranium species initially contained in the single uranium oxide particles were radioactive, the overall decay rate of the particles was very low because the uranium isotopes are long-lived radionuclides. The discussion was added to the section "3.1. Experimental results" and was removed from "3.2 Self-charging simulation of single uranium oxide particles". Then, the self-charging simulation results were discussed to assess the effects of radioactive decay on the charging of single radioactive particles.

Additional simulations have been performed to discuss the influence of radioactive decay on the charging of airborne radioactive particles. Charge and ion balance equations including self-charging and diffusion charging were used for the simulations. Based on these new results, additional discussion topics were added and the conlusions were fully revised.

**Revision:**

**2.4.2 Charging of radioactive and nonradioactive particles**

In the air, particles can acquire electric charge by capturing positive and negative ions. This charge acquisition process is called diffusion charging. While self-charging normally adds positive charge to particles, diffusion charging can lead to the accumulation of positive and negative charges on the surface of particles, indicating that both charging mechanisms can cause charging/discharging of the particles. For beta-emitting radioactive particles, the evolution of their charge distribution is given by (Clement and Harrison, 1992; Kim et al., 2015, 2016):

$$\frac{dN_j}{dt} = A_{j-1}N_{j-1} - A_j N_j + \beta_{j-1}^+ n_{ion}^+ N_{j-1} - \beta_j^+ n_{ion}^+ N_j + \beta_{j+1}^- n_{ion}^- N_{j+1} - \beta_j^- n_{ion}^- N_j$$

(6)

where $N_j$ is the concentration of particles with elementary charge $j$, $\beta_j^{\pm}$ are the ion-particle attachment coefficients, and $n_{ion}^{\pm}$ are the concentration of positive and negative ions. In Eq 6, the first two terms of the right-hand-side denote self-charging and the rest of the terms denote diffusion charging. The concentration of positive and negative ions is given as (Kim et al., 2017):

$$\frac{dn_{ion}^+}{dt} = -n_{ion}^+ \sum_j \beta_j^+ N_j - \alpha_{rc} n_{ion}^+ n_{ion}^- + S_{ion}^+$$

(7)

$$\frac{dn_{ion}^-}{dt} = -n_{ion}^- \sum_j \beta_j^- N_j - \alpha_{rc} n_{ion}^+ n_{ion}^- + S_{ion}^-$$

(8)

where $\alpha_{rc}$ is the recombination coefficient of ions and $S_{ion}^{\pm}$ are the production rate of positive and negative ions.

**3.2 Self-charging simulation of single uranium oxide particles**

Since the experimental time scale might be too short to capture UO₂ self-charging, simulations were performed to assess the effects of radioactive decay on the charging of single radioactive particles. Self-charging from radioactive decay for single UO₂ particles was predicted using the methodology described in Section 2.4. Decay simulations performed using IBIS assumed that ²³⁸U, ²³⁵U, and ²³⁴U were the only radionuclides initially present in single UO₂ particles and their mole fractions were 99.775, 0.22, and 0.005%, respectively. The decay chains for ²³⁵U, ²³⁸U, and ²³⁴U are provided in the Supplementary Information.

Figure 7 shows simulation results for self-charging of UO₂ particles with a diameter of 0.2, 0.5, 0.8, and 1.0 μm and $M_\alpha = 6$. The initial activity for each particle size was 1.01×10⁻⁹ Bq, 1.57×10⁻⁸ Bq, 6.45×10⁻⁸ Bq, and 1.25×10⁻⁷ Bq, respectively. Because of the very low radioactivity levels, The particles slowly accumulated positive charge with time due to the decay of the uranium isotopes and their daughter radionuclides such as

[234]Th and [234]Pa. Larger $UO_2$ particles gained more positive charges because the particles contained more radioactive atoms. Of the radionuclides considered in these simulations, [238]U and [234]U had the largest impact on particle self-charging, with the radionuclides together accounting for ~42.5% of the charging rate. [234]Th and [234]Pa were the next most substantial radioisotopes with the activity from each of those radionuclides accounting for about 7.1% of the predicted charge. All other radionuclides had a negligible contribution to the charging of the $UO_2$ particles. These results demonstrate the importance of each radioisotope's activity (i.e. decay frequency) on the self-charging of radioactive particles. For example, the number concentration of [238]U is approximately four orders of magnitude greater than [234]U but both iosotopes are comparable with respect to the self-charging rate. Similarly, [234]Th and [234]Pa have concentrations that are more than seven orders of magnitude smaller than [234]U but, nevertheless, remain relevant to self-charging. For $M_\alpha$ = 8, 10, and 12, the $UO_2$ self-charging effect was enhanced since each instance of alpha decay added more positive charges to the particles. These results suggest that the uranium oxide particles can spontaneously acquire positive charge via self-charging, but also that the self-charging rate is quite slow as a consequence of the decay rate of uranium.

[Figure]

**Figure 7:** Predicted average charge for $UO_2$ particles with diameters of 0.2, 0.5, 0.8, and 1.0 μm for a period of up to 730 days based on IBIS simulations of radioactive decay when $M_\alpha$ = 6.

"**4. Conclusions**
An electrodynamic balance apparatus was employed to measure the charging of various airborne dust particulates, including particles of uranium oxide, Arizona dust, urban dust, hydrophilic silica, hydrophobic silica, and graphene oxide. The charge, charge distribution, and charge density data obtained were compared to understand the triboelectric charging characteristics of the particles. The measurements showed that uranium oxide can be more easily charged by triboelectrification than the other particles. Charge-balance equations were employed to simulate the charging of triboelectrically charged radioactive and nonradioactive dust particles. Simulation results indicated that, in open air, triboelectrically charged nonradioactive particles can be quickly discharged by capturing counterions but radioactive particles can remain strongly charged because of radioactive self-charging. These results suggest that various electrostatic interactions involing radioactive particles can be created within dispersed radioactive plumes, and the effects of electrostatic interactions on dispersion of radioactive plumes should be further investigated."

12)   It seems that the presented method is not suitable to quantify the amount of particle charge induced by radioactive self-charging. For this, other techniques such as Kelvin Probe Force Microscopy seem to be better suited.

**Response**: The following discussion with reference has been added on page 6 of the revised manuscript.

"Particle levitation and surface potential microscopy have been used to investigate surface charging of single micron-size particles. Investigations using single particle levitation and surface potential microscopy are reported elsewhere (e.g., Davis and Schweiger, 2002; Kweon et al., 2013). For surface potential microscopy, sample particles should be mounted on a surface for scanning, which makes the method intrusive rather than noninstrusive. For single particle levitation, in contrast, particle mounting is not required and the net charge of single paritlces can be directly and quickly measured. Thus, in this study, we selected single particle levitation for the measurements.

Davis, E.J. and Schweiger, G., 2002. The airborne microparticle: its physics, chemistry, optics, and transport phenomena. Springer Science & Business Media.

Kweon, H., S. Yiacoumi, I. Lee, J. McFarlane, C. Tsouris, "Influence of surface potential on the adhesive force of radioactive gold surfaces," Langmuir, doi.org/10.1021/la4008476, **29**, 11876–11883 (2013).